# Drosophila miR-87 promotes dendrite regeneration by targeting the transcriptional repressor Tramtrack69

**Yasuko Kitatani**[1⊙], **Akane Tezuka**[1⊙], **Eri Hasegawa**[1⊙], **Satoyoshi Yanagi**[1], **Kazuya Togashi**[1], **Masato Tsuji**[1], **Shu Kondo**[2], **Jay Z. Parrish**[3]*, **Kazuo Emoto**[1,4]*

1 Department of Biological Sciences, Graduate School of Science, University of Tokyo, Hongo, Bunkyo-ku, Tokyo, Japan, 2 Genetic Strains Research Center, National Institute of Genetics, Yata, Mishima, Shizuoka, Japan, 3 Department of Biology, University of Washington, Seattle, Washington, United States of America, 4 International Research Center for Neurointelligence (WPI-IRCN), University of Tokyo, Hongo, Bunkyo-ku, Tokyo, Japan

⊙ These authors contributed equally to this work.

* jzp2@uw.edu (JZP); emoto@bs.s.u-tokyo.ac.jp (KE)

**Data Availability Statement:** All relevant data are within the manuscript and its supporting information files.

## Abstract

To remodel functional neuronal connectivity, neurons often alter dendrite arbors through elimination and subsequent regeneration of dendritic branches. However, the intrinsic mechanisms underlying this developmentally programmed dendrite regeneration and whether it shares common machinery with injury-induced regeneration remain largely unknown. *Drosophila* class IV dendrite arborization (C4da) sensory neurons regenerate adult-specific dendrites after eliminating larval dendrites during metamorphosis. Here we show that the microRNA *miR-87* is a critical regulator of dendrite regeneration in *Drosophila*. *miR-87* knockout impairs dendrite regeneration after developmentally-programmed pruning, whereas *miR-87* overexpression in C4da neurons leads to precocious initiation of dendrite regeneration. Genetic analyses indicate that the transcriptional repressor Tramtrack69 (Ttk69) is a functional target for *miR-87*-mediated repression as *ttk69* expression is increased in *miR-87* knockout neurons and reducing *ttk69* expression restores dendrite regeneration to mutants lacking *miR-87* function. We further show that *miR-87* is required for dendrite regeneration after acute injury in the larval stage, providing a mechanistic link between developmentally programmed and injury-induced dendrite regeneration. These findings thus indicate that *miR-87* promotes dendrite regrowth during regeneration at least in part through suppressing Ttk69 in *Drosophila* sensory neurons and suggest that developmental and injury-induced dendrite regeneration share a common intrinsic mechanism to reactivate dendrite growth.

## Author summary

Dendrites are the primary sites for synaptic and sensory inputs. To remodel or repair neuronal connectivity, dendrites often exhibit large-scale structural changes that can be triggered by developmental signals, alterations in sensory inputs, or injury. Despite the

**Funding:** This work was supported by MEXT Grants-in-Aid for Scientific Research on Innovative Areas "Dynamic regulation of brain function by Scrap & Build system" (KAKENHI 16H06456), JSPS (KAKENHI 16H02504), WPI-IRCN, AMED-CREST (JP18gm0610014), JST-CREST, the Strategic Research Program for Brain Sciences, Toray Foundation, Naito Foundation, Takeda Science Foundation, and Uehara Memorial Foundation to KE; and by grants from the National Institutes of Health (NINDS R01 NS076614) and a JSPS invitational fellowship to JZP. The funders had no role in study design, data collection and analysis, decision to publish, or preparation of the manuscript.

**Competing interests:** The authors declare that no conflicts of interest exist.

importance of dendritic remodeling to nervous system function, the molecular basis for this remodeling is largely unknown. Here we used an unbiased genetic screen and *in vivo* imaging in *Drosophila* sensory neurons to demonstrate that the microRNA *miR-87* is a critical factor required in neurons to reactivate dendritic growth both in developmental remodeling and following injury. Our work supports the model that *miR-87* promotes dendrite regeneration by blocking expression of the transcriptional repressor Tramtrack69 in neurons. This study thus establishes a role for miRNAs in temporal control of dendrite regeneration.

## Introduction

During critical periods of brain development, neurons exhibit juvenile plasticity in which connectivity can be modified in response to sensory inputs. To achieve these changes in connectivity, neurons often remodel their dendrite shape by elimination and subsequent regeneration of dendritic branches. For instance, Purkinje cells in the mouse cerebellum initially eliminate all perisomatic dendrites followed by regenerating single stem dendritic branches to form mature dendritic trees during postnatal development [1]. Likewise, during early postnatal development, layer 4 neurons in the mouse barrel cortex refine their connectivity with thalamocortical axons by biased elimination and regeneration of preexisting dendritic branches [2,3]. Over time, many types of neurons progressively reduce dynamics and stabilize their dendritic arbors as they mature [4–6]. However, dendritic arbors of mature neurons can undergo dramatic regeneration under pathological conditions such as epilepsy and after injury [7–9]. Therefore, understanding the mechanisms that underlie dendrite regeneration has important implications for understanding normal development of functional dendrite arbors and functional repair of injured neural circuits.

*Drosophila* class IV dendrite arborization (C4da) neurons exhibit both developmentally programmed and damage-induced dendrite regeneration and therefore present a genetically tractable and optically accessible model to study cellular and molecular mechanisms underlying dendrite remodeling [10–12]. During metamorphosis, dendrites which elaborate during larval stages are completely pruned away and subsequently replaced with adult-specific dendritic arbors [13–18] (Fig 1A). This developmental dendrite regeneration after pruning requires intrinsic factors including transcriptional factors [19] as well as extrinsic mechanisms such as remodeling of the extracellular matrix [17, 18]. Recent studies indicate that, in addition to this developmental dendrite regeneration, removal of a part of dendritic branches during larval stages triggers robust dendrite regeneration in C4da neurons [20–23]. In the course of injury-induced dendrite regeneration, a new dendritic process initiates growth at the severed stump by ~24 hrs after injury and then further elongates and elaborates dendritic arbors by ~72 hrs after injury [20, 21]. This progression observed in the injury-induced dendrite regeneration is morphologically similar to what has been reported during developmental dendrite regeneration, but it is unknown whether these dendrite regrowth programs share common mechanisms.

One salient feature of developmental dendrite remodeling that provides possible insight into its control is the stereotyped timing of the process, and microRNAs (miRNAs) have recently emerged as key factors regulating developmental timing in the nervous system [24–27]. miRNAs are short non-coding RNAs that interact with and generally inhibit expression of target mRNAs, typically binding target sites in the 3'UTR via base-pairing interactions and silencing gene expression via effects on mRNA stability and translation [28, 29]. Recent reports

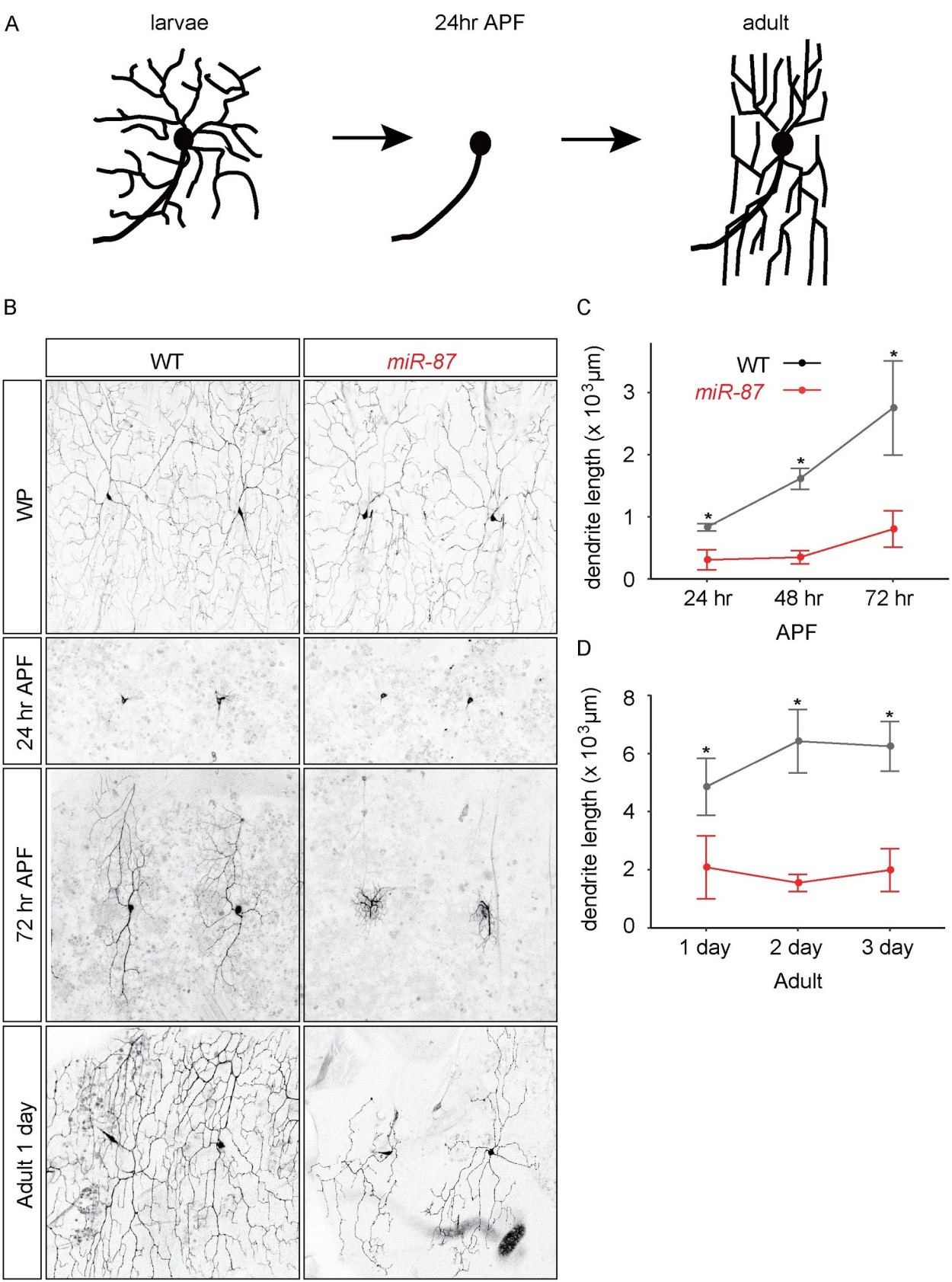

**Fig 1. *miR-87* is required for dendrite regeneration during metamorphosis.** (A) A schematic model of developmental dendrite regeneration in C4da neurons. (B) Dendrite regeneration in wild-type control (WT) and *miR-87* knockout (*miR-87*) C4da neurons at the indicated time points. WP, white pupa; APF, after pupa formation; Adult 1 day, 1 day post-eclosion. Scale bar = 100 μm. (C-D) Quantitative comparison of total dendrite length in wild-type (WT) and *miR-87* KO (*miR-87*) C4da neurons during pupal (C) and adult (D) stages. Points depict mean values, error bars indicate standard deviation values. n = 30. *p<0.01, Student's *t*-test. Genotypes: WT, *ppk-GFP*; *miR-87*, *ppk-GFP*; *miR-87^{KO}/miR-87^{KO}*.

suggest both positive and negative roles of miRNAs in axon regeneration after injury in both the central and peripheral nervous system [30–35]. For example, miR-133b is upregulated after spinal cord transection and facilitates axon regeneration probably through reducing protein level of RhoA GTPase [34]. Conversely, miR-138 suppresses mammalian axon regeneration after injury by targeting SIRT1 [35] In contrast to the accumulating evidence suggesting potential roles of miRNAs in axon regeneration, much less is known about the cell-intrinsic role of miRNAs in dendrite regeneration.

In this study, we report a crucial role of the *Drosophila* miRNA *miR-87* in dendrite regeneration of C4da neurons. Loss of *miR-87* function in C4da neurons significantly diminished dendrite regeneration, whereas *miR-87* overexpression in C4da neurons accelerated dendrite regrowth. We found that *miR-87* expression was elevated in C4da neurons during the larva-to-pupa transition, and that *miR-87* appears to promote dendrite regrowth at least in part by targeting the transcriptional repressor Tramtrack69. Finally, we found that *miR-87* is required for dendrite regeneration that occurs after injury. These data suggest that *miR-87* is a critical factor to reactivate branch outgrowth in both developmental and injury-induced dendrite regeneration.

## Results

### *miR-87* is required for dendrite regeneration during metamorphosis

To understand molecular mechanisms underlying the temporal control of dendrite regeneration, we focused on microRNAs (miRNAs) as miRNAs are implicated in regulation of developmental timing in the nervous system [24–27]. To identify miRNAs that might be involved in C4da dendrite regeneration, we systematically assayed effects of miRNA knockout using the miRNA knockout (KO) collection [36] in combination with the C4da-specific *pickpocket* (*ppk*)-GFP reporter, which is expressed in C4da neurons from late embryonic stages through adulthood [15, 37]. Among 69 autosomal miRNA KO lines we tested, covering ~80% of expressed miRNAs (S1 Table), we identified a single miRNA required for dendrite regrowth. In larvae homozygous mutant for a *miR-87* KO allele, C4da neurons showed a significant defect in dendrite degeneration during metamorphosis (Fig 1B). *miR-87* KO neurons pruned their larval dendrites normally by 24 hr APF, but failed to properly regrow dendritic trees, resulting an inappropriate coverage of the receptive fields in newly eclosed adults (Fig 1A–1C). Although wild-type C4da neurons further extended terminal branches for the first 2 days post-eclosion to completely cover the body wall, we observed no significant dendrite growth in *miR-87* KO neurons over the same period (Fig 1D, S1 Fig), suggesting that the dendrite regeneration defects in *miR-87* KO neurons are due to a reduced ability in branch regrowth, rather than a developmental delay in dendrite growth.

Similar to larval C4da dendrites, the regenerated adult C4da dendrites grow between the epidermis and the underlying musculature, largely confined to a 2-dimensonal territory [16–18]. It is thus possible that *miR-87* could function in sensory neurons, the muscle, and/or epidermal cells to control dendrite regeneration; recent studies of the miRNA *bantam* provide precedent for the latter scenario [20, 38]. To distinguish between these possibilities, we performed MARCM (mosaic analysis with a repressible cell marker) to generate single neuron

clones homozygous for *miR-87* KO in a heterozygous background [39]. Compared to wild-type controls, we found that *miR-87* MARCM clones exhibited severe dendrite regeneration defects which were comparable to the regeneration defects we observed in homozygous *miR-87* mutants (Fig 2A–2C). These defects were largely rescued by expressing *UAS-miR-87* in the *miR-87* KO clones, demonstrating that the dendrite regeneration defects were associated with loss of *miR-87* function in C4da neurons. We note that expressing *UAS-miR-87* in the *miR-87* KO clones did not fully restore dendrite length to the levels of wild-type clones; this difference may reflect differences in the timing/levels of *miR-87* expression in the rescue clones or may reflect minor cell non-autonomous contributions of *miR-87* in other tissues which are heterozygous for *miR-87* mutation. Taken together, these results demonstrate that *miR-87* is cell-autonomously required in C4da neurons for dendrite regeneration during metamorphosis.

## *miR-87* KO neurons are defective in the initial elongation of regenerating dendrites

To gain insight into the cellular basis of *miR-87* KO dendrite regeneration defects, we performed time-lapse imaging to monitor dendrite growth dynamics in wild-type and *miR-87* KO mutant larvae. Consistent with prior reports [14, 17], we found that dendritic arbors of the larval v'ada neurons were completely pruned away by ~24 hr APF in wild-type and *miR-87* KO neurons, while the soma and axonal processes remained intact (Fig 1A and 1B). We thus started our time-lapse analysis at 24 hr APF. During the first ~3 hr after the completion of dendrite arbor pruning, wild-type C4da neurons elaborated dynamic protrusions which extended and retracted from the soma. These fine branches were transient structures (S1 Movie), therefore C4da neurons exhibited very little net regrowth of dendritic branches during this time window which we refer to as the "pausing" stage of dendrite regrowth. Next, 1–2 branches among the many existing protrusions thickened at ~27 hr APF and subsequently elongated, eventually forming major dendrite branches by ~36 hr APF that would serve as the scaffold of adult C4da dendrite arbors (Fig 3, S1 Movie). C4da neurons in *miR-87* KO mutants similarly elaborated fine, dynamic protrusions and exhibited thickening of select branches at ~27 hr APF. However, unlike wild-type controls, the nascent "thick" branches failed to elongate in *miR-87* KO mutants. Consequently, dendrite regrowth was largely arrested (Fig 3, S2 Movie). We noted that the nascent dendritic branches of *miR-87* neurons made frequent contact with one another during this window of dendrite regrowth, however contact events were not followed by dendrite retraction, suggesting that dendrite regeneration defects in *miR-87* neurons are unrelated to contact-dependent retraction of dendritic branches. These observations suggest that *miR-87* is required for efficient elongation of dendritic branches during regeneration in C4da neurons.

## *miR-87* activity is elevated in C4da neurons during the larva-to-pupa transition

Given the temporal requirement for *miR-87* for dendrite regrowth after ~27 hr APF, we next investigated whether *miR-87* activity was induced to trigger regrowth. A previous northern blot analysis showed that *miR-87* expression in whole animals is markedly increased during the larva-to-pupa transition [40], however the dynamics of *miR-87* expression in C4da neurons have not been documented. To directly monitor *miR-87* activity in C4da neurons, we generated a *miR-87* activity sensor that contains four *miR-87* binding sites in the 3′UTR of GFP, hence GFP expression is attenuated in cells expressing mature *miR-87* (Fig 4A) [41]. As a control, we first monitored expression of a control GFP sensor that lacked *miR-87* binding sites and therefore should be refractory to *miR-87* expression levels. We found that this control

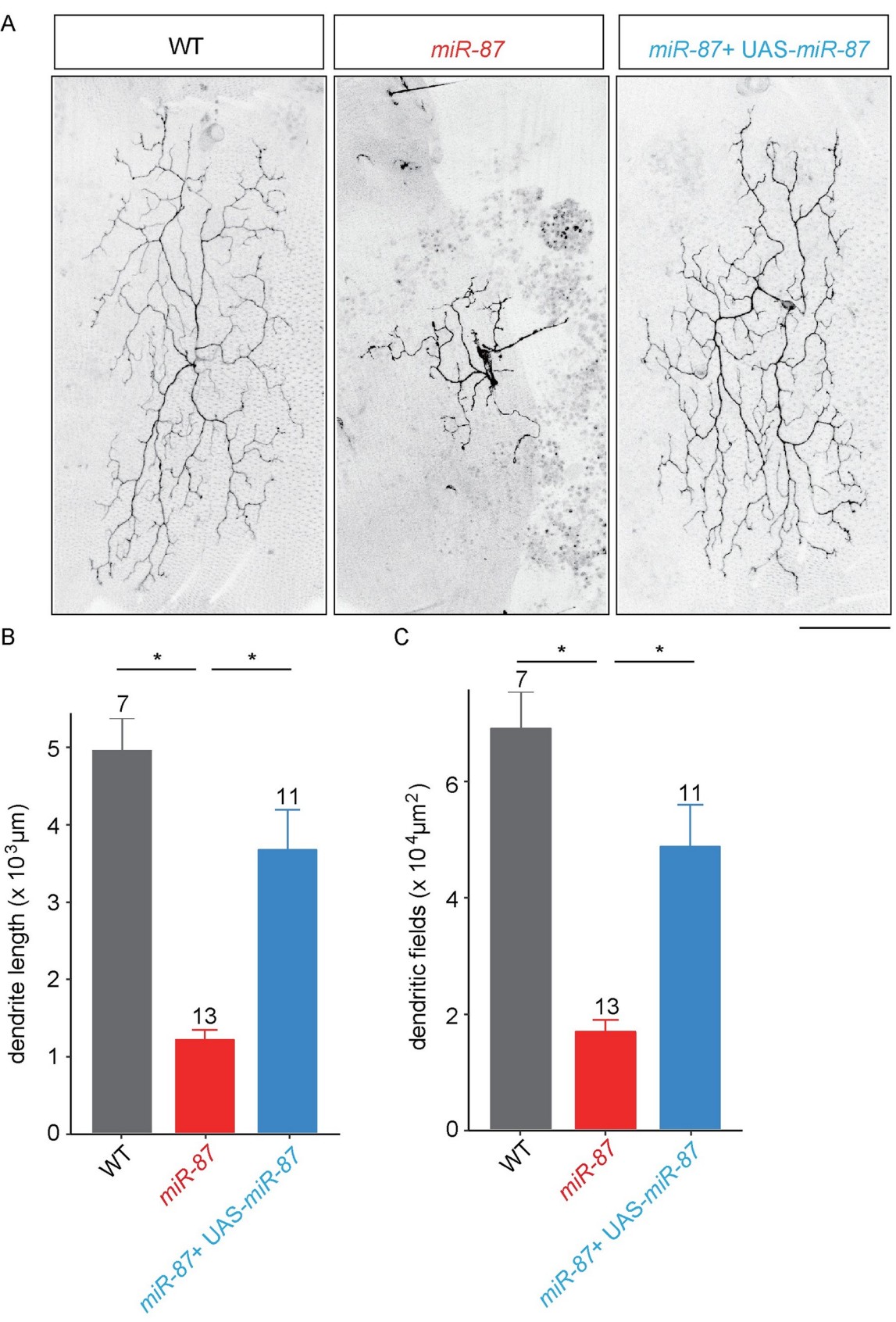

**Fig 2. *miR-87* functions cell-autonomously in C4da neurons to promote dendrite regeneration.** (A) MARCM analysis of wild-type control (WT), *miR-87* KO (*miR-87*), and *miR-87* KO with UAS-*miR-87* expression (*miR-87* + UAS-*miR-87*) C4da neurons at 96 hrs APF. Scale bar, 100 μm. (B, C) Quantitative analysis of the total dendrite length (B) and the dendritic field areas (C) of MARCM clones. The sample numbers examined are shown on the top of each column, Bars indicate mean ± standard deviation, *p<0.01 (ANOVA with a post hoc Bonferroni correction). Clone genotypes: (WT) *hsFLP, ppk-Gal4, UAS-mCD8GFP/+; FRT40A*, (*miR-87*) *hsFLP, ppk-Gal4, UAS-mCD8GFP/+; miR-87, FRT40A*, and (*miR-87* + UAS-*miR-87*) *hsFLP, ppk-Gal4, UAS-mCD8GFP/+; miR-87, FRT40A; UAS-miR-87/+*.

sensor supported uniform levels of GFP expression in C4da neurons during larval and early pupal stages (Fig 4B and 4C and 4H). In contrast, whereas the *miR-87* sensor produced high levels of GFP expression in C4da neurons during early larval stages, we observed significant attenuation of GFP expression by the *miR-87* sensor beginning in 3rd instar larvae, resulting in a >50% reduction of GFP expression in early pupal stages (Fig 4D–4H). Notably, the temporal control of the *miR-87* sensor was dependent on *miR-87* activity, as GFP expression by the *miR-87* sensor was not attenuated in *miR-87* mutant C4da neurons (Fig 4I). Thus, *miR-87* activity in C4da neurons is increased during the larva-to-pupa transition, in anticipation of *miR-87* function in dendrite regrowth.

Given that *miR-87* activity is upregulated and required for dendrite regeneration, we reasoned that elevation of *miR-87* activity might promote C4da dendrite regeneration. To examine this possibility, we constitutively overexpressed *miR-87* in C4da neurons from early developmental stages using the C4da-specific *ppk-Gal4* driver. As described above, dendrite pruning is complete at 24 hr APF in wild-type and *miR-87* KO C4da neurons (Fig 1B and 1C). We found that dendrite pruning was completed on a similar timescale in *miR-87*-overexpressing neurons (S2 Fig), however *miR-87* overexpression led to precocious dendrite outgrowth following pruning. First, whereas control C4da neurons exhibited short, fine protrusions at 24 hr APF, the length of each protrusion was significantly increased in *miR-87*-overexpressing

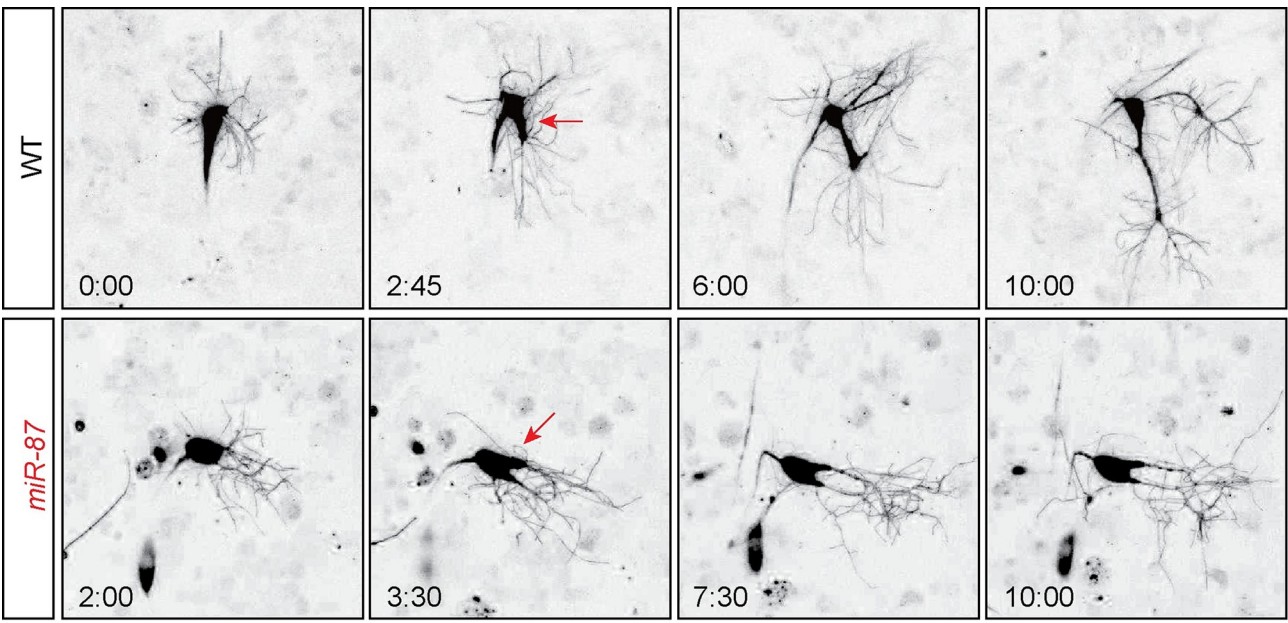

**Fig 3. *miR-87* is required for branch elongation in dendrite regeneration.** Time-lapse imaging of dendrite regeneration in wild-type control (WT) and *miR-87* KO (*miR-87*) C4da neurons. Recording started at 24 hr APF and continued for ~12 hrs. Time stamps represent the intervals after 24 hr APF (hours: minutes). Red arrows indicate branches undergoing thickening. Scale bar = 25μm. Genotypes: WT, *ppk-GFP*; *miR-87*, *ppk-GFP*; *miR-87^{KO}/miR-87^{KO}*.

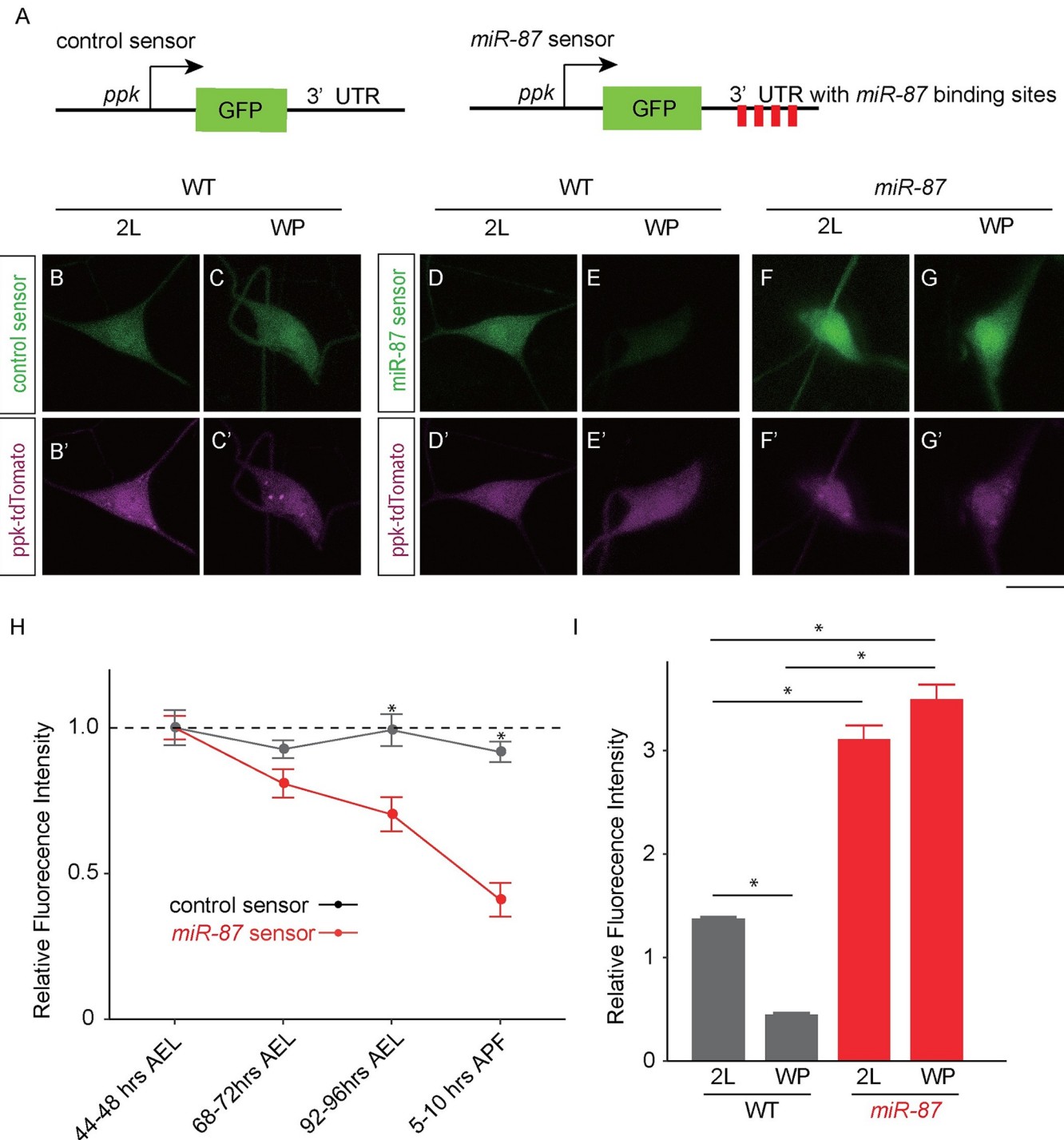

**Fig 4. *miR-87* expression is elevated in C4da neurons during the larvae-to-pupae transition.** (A) Schematic representations of control and *miR-87* sensor constructs. (B-C) Control sensor is stably expressed in C4da neurons. Control sensor expression in animals additionally expressing *ppk*-CD4-tdTomato is shown in 2nd instar larval (2L) and white pupal (WP) C4da neurons. (D-G) Regulated expression of *miR-87* sensor. Images show expression of the *miR-87* sensor in 2nd instar larval (2L) and white pupal (WP) C4da neurons additionally expressing *ppk*-CD4-tdTomato. Note the marked downregulation of GFP expression over this interval. (F, G) *miR-87* sensor expression is responsive to *miR-87* activity. Image depicts *miR-87* sensor expression in 2L and WP C4da neurons of *miR-87* KO animals. Sensor expression remains high in *miR-87* KO neurons, indicating that pupal downregulation of the sensor depends on *miR-87* function. Scale bar = 20 μm. (H) Quantification of control and *miR-87* sensor GFP expression in C4da neurons at the indicated developmental stages. AEL, after egg laying. The *miR-87* sensor fluorescence intensity was normalized to the control sensor fluorescence intensity and represented as relative intensity to 2nd instar larval levels (44–48 hr AFL). n = 25, Error bars indicate mean ± S.D, *p<0.001 (Student's *t*-test). (I) Quantification of relative fluorescence intensity of *miR-87* sensor GFP in wild-type (WT) and *miR-87* KO (*miR-87*) C4da neurons. The *miR-87* sensor fluorescence intensity was normalized to the

fluorescence intensity of the control sensor lacking *miR-87* binding sites in the same genetic background and at the same developmental stage. n = 25, Error bars indicate mean ± S.D, *p<0.01 (Student's *t*-test). Genotypes: control sensor in WT neurons: *ppk::tdTomato; +/+; ppk-miR-87-control sensor; miR-87* sensor in WT neurons, *ppk::tdTomato; +/+; ppk-miR-87 sensor; miR-87* sensor in *miR-87* neurons, *ppk::tdTomato; miR-87$^{KO}$/miR-87$^{KO}$; ppk-miR-87 sensor.*

neurons (Fig 5A and 5B; wild-type, 12.4 ± 0.5 μm; *miR-87* o/e, 35.2 ± 1.2 μm, p< 0.01, n = 30 protrusions from 10 different neurons). Second, whereas control neurons remained in the "pausing" phase of regrowth at 30 hr APF, with control dendrites exhibiting no significant net dendrite growth, *miR-87*-overexpressing neurons had extended 2–3 thick branches by 30 hr APF (Fig 5A and 5C; wild-type, 0.3 ± 0.1 branches; *miR-87* o/e, 2.7 ± 0.6 branches, p< 0.01, n = 30). Finally, when we monitored the status of dendrite regrowth at 72 hr APF, we found that both wild type and *miR-87*-overexpressing neurons had regrown to a comparable degree (Fig 5A and 5D), demonstrating that *miR-87* overexpression specifically regulates the timing of early stages of regrowth, not the overall extent of regrowth. These results together support the idea that elevation of the *miR-87* expression during the larvae-to-pupae transition triggers dendrite regeneration in C4da neurons.

## miR-87 targets the transcriptional repressor Tramtrack69 to promote dendrite regeneration

miRNAs typically exert their control via base-pairing interactions with 3'UTR sequences in their targets, leading to translational repression [28, 29]. These base-pairing interactions allow for computational prediction of miRNA targets; such a prediction has identified more than 100 putative *miR-87* targets [42]. As a first step to determine the functionally relevant targets of *miR-87* in C4da dendrite regeneration, we used gain-of-function approaches, reasoning that overexpression of bona fide *miR-87* targets in a wild-type background should phenocopy loss of *miR-87* function. Indeed, among 43 potential targets we tested (S2 Table), overexpression of 3 genes, *hepatocyte nuclear factor 4* (*hnf4*), *broad* (*br*), and *tramtrack69* (*ttk69*), caused significant defects in C4da dendrite regeneration (Fig 6A and 6B, S3 Fig, S5A and S5B Fig).

To further define the functional relevance of these putative *miR-87* targets, we next tested for dosage-sensitive interactions between the candidate genes and *miR-87*, assaying whether heterozygous loss-of-function mutations in the target genes could suppress the dendrite regeneration defect in *miR-87* KO neurons. Strikingly, removal of one copy of *ttk69* significantly restored dendrite growth during regeneration in *miR-87* KO animals whereas removal of one copy of *hnf4* or *br* caused no significant rescue in the dendrite regeneration defect in *miR-87* KO neurons (Fig 6C and 6D, S3B and S3C Fig). Similarly, targeted expression of UAS-*ttk69*-RNAi in C4da neurons partially rescued dendrite regrowth in *miR-87* KO neurons (Fig 6D), demonstrating that *ttk69* cell-autonomously regulates C4da dendrite regeneration. These results strongly suggest that Ttk69 is a key target of *miR-87* in C4da dendrite regeneration, and we further tested this possibility by assaying effects of *miR-87* on nervous system expression of *ttk69* using qPCR. In wild-type control larvae, we found that the level of *ttk69* mRNA was reduced during the larva-to-pupa transition, but this reduction was not observed in *miR-87* KO animals (Fig 6E), consistent with a role for *miR-87* in control of *ttk69* expression.

Alterations in *ttk69* levels could reflect transcriptional or post-transcriptional regulation of gene expression. To discriminate between these possibilities, we assayed whether *miR-87* affected expression of a ubiquitously expressed GFP sensor containing the *ttk69* 3'UTR (S4A Fig), which was previously shown to mediate miRNA-dependent post-transcriptional control of *ttk69* expression [43]. We found that *ttk69* GFP sensor expression was under developmental control in C4da neurons, with sensor expression significantly reduced during the larva-to-pupa transition in wild-type controls (S4B and S4C Fig). In contrast, this developmental

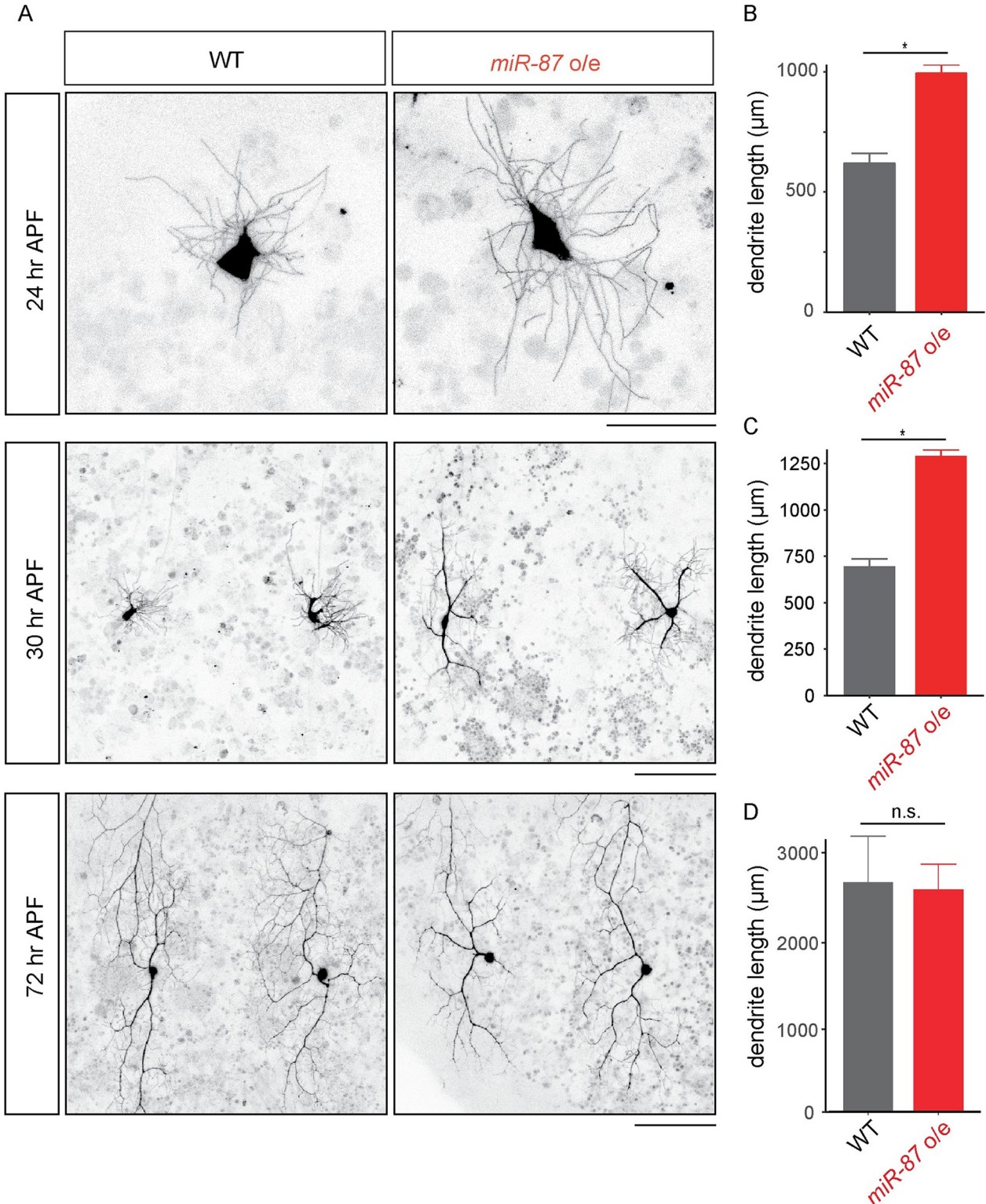

**Fig 5. *miR-87* overexpression in C4da neurons causes precocious dendrite regeneration.** (A) Live-imaging of wild-type (WT) and *miR-87*-overexpressing (*miR-87 o/e*) C4da dendrites. Scale bars = 20 μm for 24 hr APF, and 100 μm for 30hr and 72 hr APF. (B-D) Quantification of the total dendrite length in wild-type (WT) and *miR-87*-overexpressing (*miR-87 o/e*) C4da dendrites at 24 hr (B), 30 hr (C), and 72 hr (D) APF. n = 30, Error bars indicate mean ± S.D., *p<0.01 (Student's *t*-test), n.s., not significant. Genotypes: WT, *ppk-Gal4, UAS-mCD8GFP*; *miR-87o/e, ppk-GAL4, UAS-mCD8GFP; UAS-miR-87*.

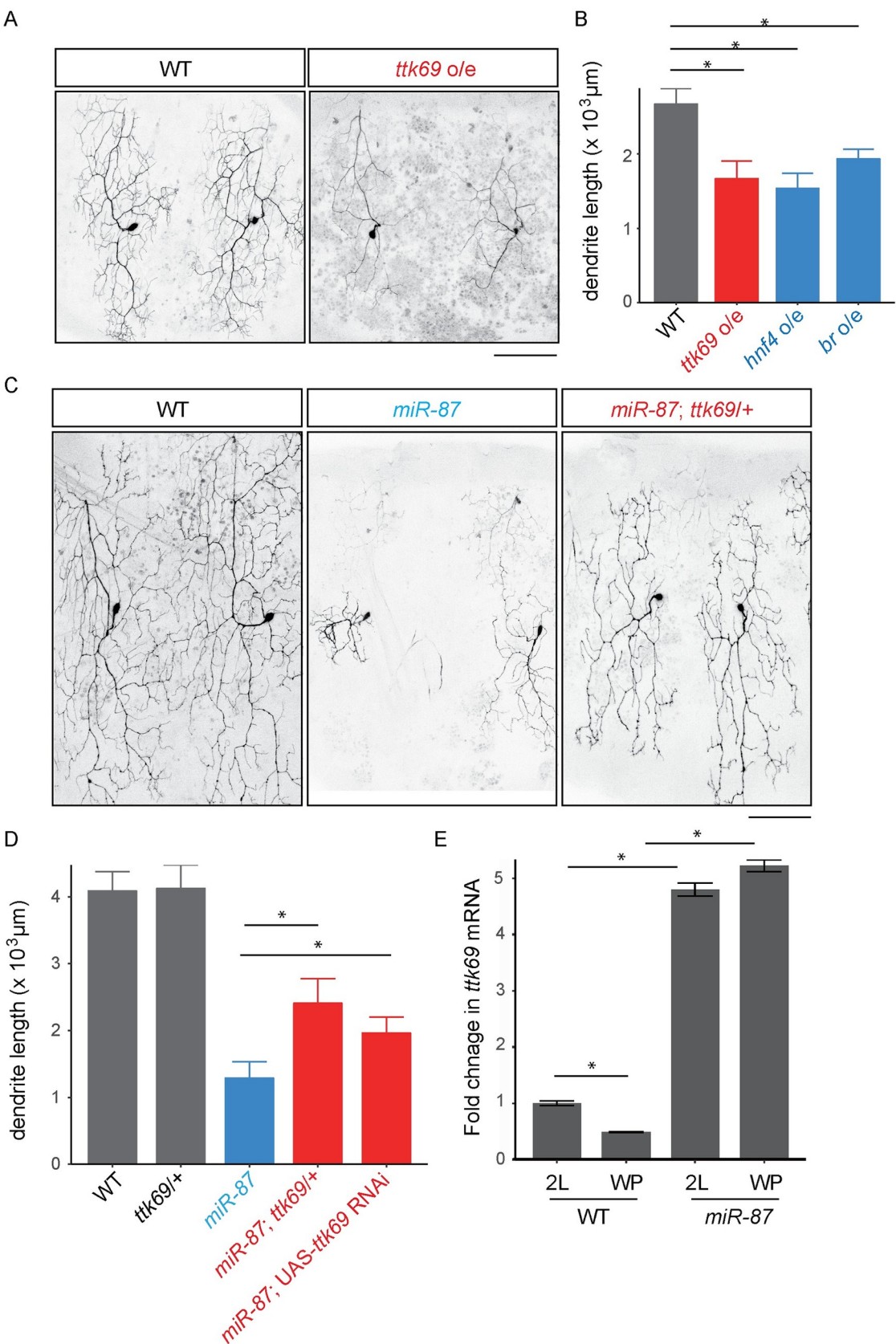

**Fig 6. *miR-87* targets the transcriptional repressor Tramtrack69 to promote dendrite regeneration.** (A, B) Overexpression of *ttk69* causes dendrite regeneration defects. Morphology of wild-type control (WT) and *ttk69* overexpressing (*ttk69* o/e) dendrites at 72 hr APF (A). Scale bar = 100 μm. Quantification of total dendrite length in wild-type (WT), *ttk69* overexpressing (*ttk69* o/e), *hnf4* overexpressing (*fnf4* o/e), and *br* overexpressing (*br* o/e) neurons (B). n = 15, Error bars indicate mean ± S.D., *p<0.01 (ANOVA with a post hoc Bonferroni correction). (C, D) Reduction of *ttk69* dosage significantly rescues dendrite regeneration defects in *miR-87* KO C4da neurons. Morphology of control wild-type (WT), *miR-87* KO (*miR-87*), and *miR-87* KO with reduction of one copy of *ttk69* (*miR-87*; *ttk69+/-*) dendrites at 96 hr APF (C). Scale bar = 100 μm. Quantification of total dendrite length in control wild-type (WT), *miR-87* KO (*miR-87*), *miR-87* KO with reduction of one copy of *ttk69* (*miR-87*; *ttk69+/-*), and *miR-87* KO with *ttk69* RNAi (*miR-87*; *UAS-ttk69* RNAi) dendrites at 96 hr APF (D). n = 15, Error bars indicate mean ± S.D., *p<0.01 (ANOVA with a post hoc Bonferroni correction). (E) Quantitative PCR showing *ttk69* mRNA levels from brain extracts of the indicated genotypes at second instar (2L) or white pupal (WP) stages. Data were normalized to *rp49* and represent the average of 5 independent experiments. Error bars indicate mean ± S.D., *p<0.01, (Student's *t*-test). Genotypes: WT, *ppk-Gal4, UAS-mCD8GFP; ttk69* o/e, *ppk-GAL4, UAS-mCD8GFP; UAS-ttk69/+; miR-87, ppk-Gal4, UAS-mCD8GFP; miR-87^{KO}/ miR-87^{KO}; miR-87; ttk69/+, ppk-Gal4, UAS-mCD8GFP; miR-87^{KO}/ miR-87^{KO}; ttk^{1e11}/+.*

control of *ttk69* sensor expression was lost in *mir-87* KO mutants, which exhibited sustained *ttk69* sensor expression at significantly higher levels than in wild type controls. Altogether, these results suggest that *mir-87* repression of *ttk69* expression is mediated through the *ttk69* 3'UTR. Taken together with genetic interaction studies, these findings suggest that *miR-87* function in dendrite regeneration depends in part on control of *ttk69* expression in C4da neurons.

## *miR-87* is required for injury-induced dendrite regeneration

Previous studies indicate that larval C4da neurons are capable of regenerating dendrites after injury [20–22]. To determine whether *miR-87* might also play a role in injury-induced dendrite regeneration, we assayed injury-induced dendrite regeneration in *miR-87* KO larvae. To this end, we severed dendrites of individual C4da neurons using a two-photon laser at ~48 hr AEL, with the lesion site restricted to the primary dendritic branch point, and monitored dendrite regeneration using time-lapse confocal microscopy (Fig 7A). Consistent with previous studies, we observed robust dendrite regrowth in wild-type larvae, with significant new growth evident at 12 hr after lesioning and dendrite regrowth continuing through the end of our imaging paradigm, 54 hr after lesioning (Fig 7A and 7B). In contrast, we observed no sign of dendrite regeneration in *miR-87* KO C4da neurons at either early or late time points after laser severing, suggesting that *miR-87* plays an essential role in injury-induced C4da dendrite regeneration (Fig 7A and 7B). We found that these degeneration defects in *miR-87* KO neurons were rescued to wild-type levels of regeneration by C4da neuron-specific expression of *miR-87*. Further, overexpression of *miR-87* in C4da neurons tended to enhance dendrite regeneration after injury (Fig 7B; WT, 746.71 ± 190.13 μm; WT + *UAS-miR87*, 1101.13 ± 201.33 μm). These data suggest that *miR-87* promotes larval dendrite regeneration following injury in a cell-autonomous manner. Similar to developmental dendrite regeneration, *ttk69* knockdown significantly enhanced injury-induced dendrite regrowth in *miR-87* mutant C4da neurons (Fig 7B), whereas overexpression of *ttk69* in C4da neurons caused dendrite regrowth defects in otherwise wild-type C4da neurons (S5C and S5D Fig). These data suggest that, similar to developmentally programmed dendrite regrowth, dendrite regrowth after injury critically depends on *miR-87* control of a gene expression program that involves *ttk69*.

Finally, to test whether dendrite injury induces expression of this regeneration program, we monitored effects of dendrite severing on *miR-87* sensor expression in C4da neurons. Indeed, *miR-87* sensor fluorescent intensity was significantly reduced 12 hr after dendrite severing in control but not *miR-87* mutant larvae, suggesting that dendrite damage induced *miR-87* activity in C4da neurons (Fig 7C and 7D). Altogether, these data suggest that *miR-87* plays a central role in dendrite regrowth, potentiating dendrite regrowth following both injury-induced

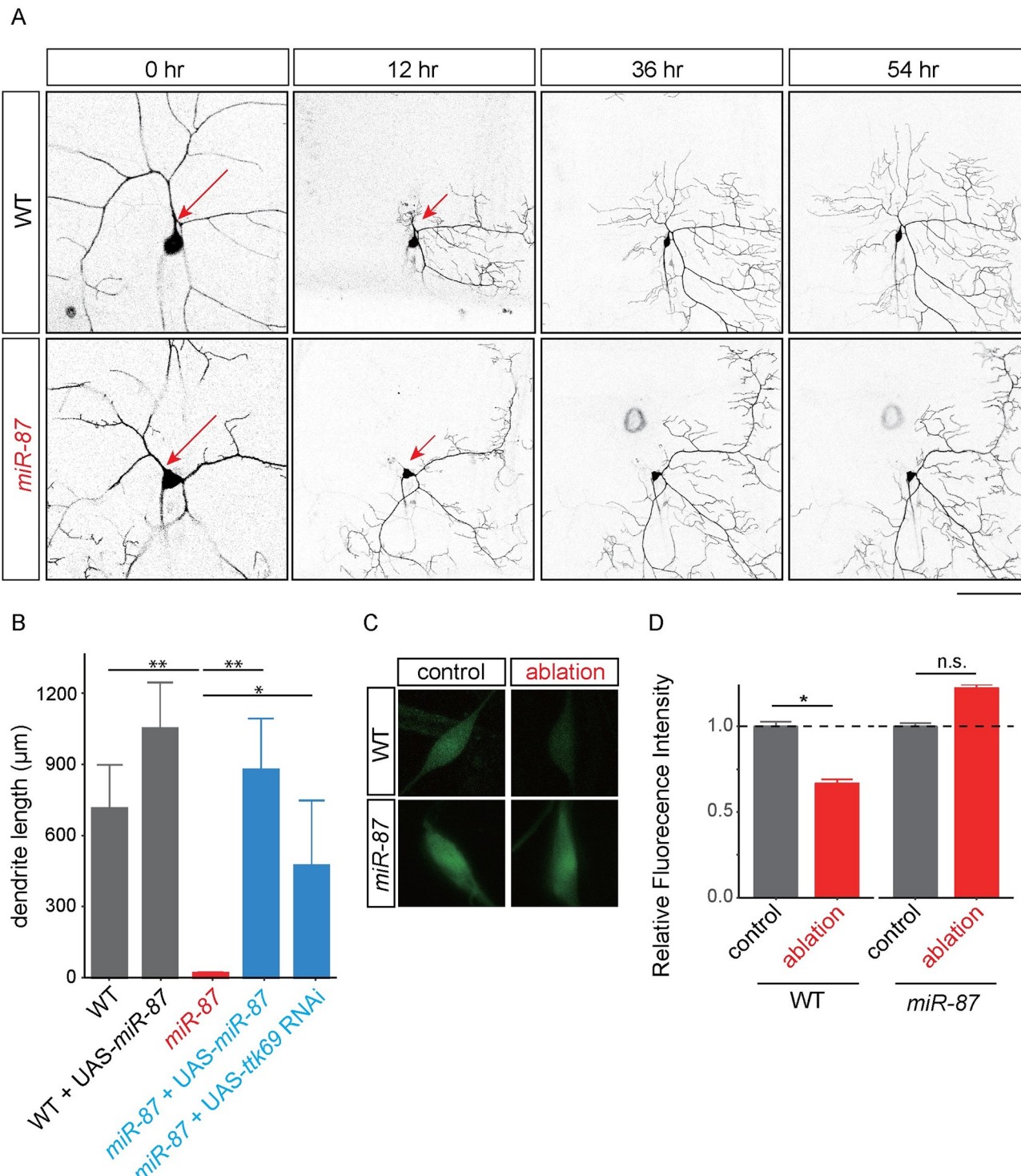

**Fig 7. *miR-87* is required for injury-induced dendrite regeneration.** (A) Time-lapse images of dendrite regeneration after injury in wild-type (WT) and *miR-87* KO (*miR-87*) neurons. Red arrows indicate the site of dendrite severing in 2nd instar larvae (44–48 hr AEL). Scale bar = 25 μm and 50 μm for 0 hr and 12–54 hrs after severing, respectively. (B) Quantification of regenerated dendrite length after severing in control wild-type (WT), wild-type overexpressing *miR-87* (WT + *UAS-miR-87*) *miR-87* KO (*miR-87*), *miR-87* KO with C4da-specific rescue of *miR-87* (*miR-87 + UAS-miR-87*), and *miR-87* KO with *ttk69* RNAi (*miR-87*; UAS-*ttk69* RNAi) dendrites at 54 hr after severing. n = 25, Error bars indicate mean ± S.D., \*\*p<0.01, \*p<0.05 (ANOVA with a post hoc Bonferroni correction). (C) Expression of *miR-87* sensor in control and injured neurons showed a marked down-regulation of GFP expression 12 hrs after dendrite severing. *miR-87* sensor expression remains high in *miR-87* KO neurons after dendrite severing, indicating that the downregulation depends on *miR-87* function. Scale bar = 20 μm. (D) Quantification of *miR-87* sensor GFP in control and injured C4da neurons. The

*miR-87* sensor fluorescence intensity was normalized to the control sensor fluorescence intensity in the same genetic background and at the same developmental stage. n = 15 neurons from independent animals for each genotype/condition, Error bars indicate mean ± S.D., *p<0.01 (unpaired *t*-test), n.s., not significant. Genotypes: WT, *ppk-Gal4, UAS-mCD8GFP*; WT + *UAS-miR-87, ppk-GAL4, UAS-mCD8GFP; UAS-miR-87/+*; *miR-87, ppk-Gal4, UAS-mCD8GFP; miR-87^{KO}/ miR-87^{KO}*; *miR-87 + UAS-miR-87, ppk-Gal4, UAS-mCD8GFP; miR-87^{KO}/ miR-87^{KO}; UAS-miR-87/+*; *miR-87 + UAS-ttk69 RNAi, ppk-Gal4, UAS-mCD8GFP; miR-87^{KO}/ miR-87^{KO}; UAS-ttk69 RNAi /+*.

dendrite regeneration and developmental dendrite regeneration by suppressing expression of the transcription factor Ttk69.

## Discussion

In this study, through a non-biased miRNA screen in *Drosophila* sensory neurons, we have for the first time identified the miRNA *miR-87* as a critical regulator of dendrite regeneration. Despite the many links between miRNAs and neuronal development, including axon regeneration [30–35], these studies provide one of the first connections between miRNAs and dendrite regeneration. In *Drosophila*, *miR-87* KO C4da neurons fail to regenerate dendritic arbors, resulting in incomplete coverage of the adult epidermis (Fig 1). These defects are ameliorated substantially by expression of *miR-87* in KO C4da neurons (Fig 2), indicating that *miR-87* functions in a cell-autonomous manner to regulate dendrite regeneration. Furthermore, our time-lapse analysis showed that *miR-87* mutant C4da neurons are defective in the initial elongation of dendritic branches (Fig 3, S1 & S2 Movies), suggesting that *miR-87* is required for pruned dendrites to reinitiate the developmental program for dendrite outgrowth. Given that *miR-87* activity in C4da neurons is elevated during the larvae-to-pupae transition (Fig 4), we propose a model in which upregulation of *miR-87* is a critical determinant for dendrite regrowth. In support of this model, forced expression of *miR-87* in C4da neurons in the early larval stages causes precocious initiation of dendrite regeneration (Fig 5).

How could *miR-87* promote dendrite regeneration in C4da neurons? Given that miRNAs typically bind to 3'UTR of particular mRNAs and suppress their expression [28, 29], the most likely scenario is that *miR-87* promotes dendrite regeneration by suppressing target gene expression in C4da neurons. Our genetic and biochemical evidence indicates that the transcriptional repressor Ttk69 is a key functional target for *miR-87* in dendrite regeneration. Overexpression of *ttk69* in C4da neurons leads to dendrite regeneration defects (Fig 6). Further, both heterozygosity for *ttk69* and RNAi knockdown of *ttk69* significantly rescue the dendrite regeneration defects in *miR-87* KO C4da neurons (Fig 6). Our qPCR data demonstrate that *ttk69* mRNA levels are developmentally regulated in the nervous system and subject to regulation by *miR-87*. Intriguingly, *ttk69* mRNA levels are reduced in the larvae-to-pupae transition and significantly increased in *miR-87* mutants (Fig 6), suggesting that the increased activity of *miR-87* during the larvae-to-pupae transition could suppress pupal *ttk69* expression. Whether this reciprocal expression relationship reflects direct control of *ttk69* expression by *miR-87* remains to be determined. And while our results support a key role for Ttk69 in dendrite regeneration, we cannot rule out the possibility that additional *miR-87* targets contribute to regeneration. Further studies comparing gene expression between wild-type and *miR-87* KO C4da neurons might help to identify additional *miR-87* targets in dendrite regeneration.

Although functions for Ttk69 in C4da dendrite morphogenesis or regeneration have not been previously described, prior studies provide some insights into likely control of dendrite regeneration by Ttk69. In the developing eye and the embryonic PNS, Ttk69 prevents progenitor cell differentiation by suppressing expression of genes required for neural fate specification [44–47]. In the context of regeneration, Ttk69 could be controlling expression of an adult C4da neuron "differentiation" program required for dendrite regrowth. In this model, *miR-87*

repression of *ttk69* in pupal C4da neurons would facilitate differentiation. One key feature of this model is the precise temporal control; dendrite regrowth occurs during a stereotyped developmental window, with *miR-87* required for neurons to transition from the "pausing" to elongation phases of dendrite regrowth. Indeed, the few studies of Ttk69 function in post-mitotic neurons demonstrate roles for Ttk69 in precisely timed developmental events. For example, during embryonic development of C1da dendrite arbors, *ttk69* depletion prevents the timely transition from dorsal outgrowth to lateral branching [48]. Ttk69 likewise regulates a late event of R7 photoreceptor axon targeting, namely bouton formation after axons reach the medulla [49, 50].

How is this timing achieved? As in dendrite regeneration, many of the other precisely timed Ttk69 functions are gated by miRNA control including *miR-7* in oogenesis, *miR-184* in blastoderm embryos, and *miR-310* cluster miRNAs under conditions of nutrient stress [43, 51–52], suggesting that Ttk69 may be particularly susceptible to miRNA control. The question therefore becomes: how is the temporal activity of miRNAs that regulate *ttk69* gated? Developmental timing in insects is controlled by several hormonal cues, most notably including ecdysone, with spikes in ecdysone titer preceding and required for major developmental transitions [53]. Intriguingly, ecdysone titers drop at the onset of pupal development, and then peak again at ~24 h APF [54], concomitant with the timing of dendrite regrowth. It therefore seems likely that *miR-87* and hence *ttk69* expression dynamics are controlled in part by the larval ecdysone pulse that triggers pupation and/or the early pupal pulse that precedes dendrite regrowth. Studies in S2 cells suggest that a direct connection between ecdysone and *miR-87* is unlikely [40]. Instead, ecdysone-responsive genes that are absent in S2 cells likely induce *miR-87* expression to trigger dendrite regrowth. Indeed, temporal specificity of ecdysone signaling is imparted by stage-specific induction of ecdysone-responsive transcription factors including Eip75B, HR3, and HR4, each of which are induced by the late larval ecdysone pulse [55] and absent or lowly expressed in S2 cells [56].

Our data demonstrate that *miR-87* is required for dendrite regeneration after acute injury as well as during developmental remodeling (Fig 7). We thus propose that the *miR-87*-mediated Ttk69 downregulation might be a core mechanism to reactivate dendrite regrowth in both developmental and injury induced regeneration in C4da neurons. Our observation that *miR-87* activity is induced by dendrite damage raises several significant questions with far-reaching implications for dendrite regeneration. What controls *miR-87* activity in sensory neurons? How are dendrite injuries sensed and transduced to induce *miR-87* activation? The latter questions are particularly significant, and the single cell resolution provided by our *miR-87* sensor should allow for systematic characterization of the mechanisms involved in damage sensing and signal transduction.

Our findings suggest that modulating miRNA activity in neurons might be a potential therapeutic strategy for promoting dendrite regeneration and functional repair after nervous system damage. Recent studies indicate that expression of a variety of miRNAs are induced following traumatic brain injury, spinal cord injury, and peripheral nerve injury [30, 31]; the physiological roles of these miRNAs in injury responses remain to be determined.

## Materials and methods

### Fly stocks

*Ppk-GFP* reporter and *Ppk-Gal4* were used in previous studies [5, 16]. *miR-87^KO^*, *ttk69^1e11^*, *hnf4^Δ33^*, *br^npr-3^*, *UAS-ttk69*, *UAS-br*, *UAS-hnf4*, *UAS-ttk69 RNAi* were obtained from Bloomington Stock Center. *UAS-miR-87* was obtained from FlyORF. The *ttk69* 3'UTR sensor stock was obtained from Dr. Deng (Florida State University).

### *miR-87* sensor

We designed our *miR-87* sensor following a previous report [41]. In brief, we generated P-element-based sensors containing an unmodified 3'UTR or a 3'UTR with high-affinity *miR-87* binding sites, and then shuttled the sensors into an attB-containing plasmid containing the C4da-specific *ppk*-promoter. First, four copies of the complementary sequence of the putative *miR-87* targeting sequence (TCACACACCTGAAATTTTGCTCAA) were cloned downstream of EGFP in the previously described miRNA sensor construct, a modified pCaSpeR4 vector with tub-EGFP inserted in 3' end of the P element [41]. The resulting pCaSpeR4-tub-EGFP with 4 x *miR-87* target sequences was used as a template to amplify EGFP (for control sensor) and EGFP with 4 x *miR-87* targets (for *miR-87* sensor) fragments. The following primers were used: for control sensor, 5'-GGTACCAACTTAAAAAAAAAAATCAAAATGGTGAGCAAG GGCGAGGA and 5'- CTCGAGTTACTTGTACAGCTCGTCCA; for *miR-87* sensor, 5'-GGT ACCAACTTAAAAAAAAAAATCAAAATGGTGAGCAAGGGCGAGGA and 5'- CTCGAG TCCGGTTGAGCAAA. PCR products were cloned into pENTR (Thermo Fisher Scientific) as a KpnI/XhoI fragment. Using Gateway LR reaction (Thermo Fisher Scientific), EGFP and EGFP with 4 x *miR-87* targets were recombined into the pDEST-APPHIH vector (a gift from Chun Han) to generate the control (*ppk*-EGFP-UTR) and *mir-87* (*ppk*-EGFP-*miR-87* binding sites-UTR) sensors used in these studies. Transgenic flies with the construct integrated at the attP2 landing site were generated by BestGene Inc.

### MARCM analysis

For MARCM analyses of *miR-87* KO C4da neurons, *FRT40A* (control) and *miR-87^{KO}*, *FRT40A* flies were crossed to *ppk-Gal4, UAS-mCD8GFP, hs-flp; tub-Gal80, FRT40A* flies. Embryos were collected for 2 hr at 25°C and allowed to develop for 3 hr in yeasted agar plates, then heat shocked for 1 hr at 37°C. Heat-shocked embryos were kept at 25°C and C4da MARCM clones were analyzed at third instar larvae. The animals with clones were further aged until the desired developmental stage.

### *In vivo* imaging

Dendrite regeneration in C4da neurons during the pupa-to-adult stages was live-imaged by confocal microscopy (Leica SP8) using a prep previously described [15, 37].

### Quantitative RT-PCR

mRNA preparation from larval and pupal brains and qRT-PCR were performed as previously described [38]. In brief, 2nd instar larvae (40–48 AEL) and 24 hr APF pupa were dissected and brains were extracted in cold PBS. Total RNA was extracted using a RNAqueous-Micro Total RNA Isolation Kit (Invitrogen) from 50–70 larval brains and 20–30 pupal brains, respectively. First-strand cDNA was synthesized using ReverTra Ace qPCR RT Master Mix (Toyobo). qRT-PCR was performed using THUNDERBIRD SYBR qPCR Mix (Toyobo). Primers used to assay *ttk69* RNA levels were F5'-ATCAAAGAACTCCAAGGATCACCG-3' and R 5'-ATGA TGTGTCCAGACCTTCGC-3'. Measurements were normalized to ribosomal protein 49 (*rp49*: F 5'-GCTAAGCTGTCGCACAAA-3' and R 5'-TCCGGTGGGCAGCATGTG-3'). Data were analyzed by the Pfaffl method.

### Dendrite lesion

A 2nd instar larva (48–52 hr AEL) was mounted on a coverslip, and single primary branches of C4da dendrites (~50 μm away from the soma) were targeted using a focused 900 nm two

photon laser (Leica MP8). Following the lesion, animals were recovered on yeasted apple juice agar plates and live-imaged at appropriate stages.

## Quantification

For quantification, C4da neurons in segments A3-5 were used. Dendrites of C4da neurons were traced using the ImageJ plug-in NeuroJ (NIH, Bethesda, MD). Total dendritic length was calculated from the traces. Dendritic fields were calculated by measuring the area bounded by a polygon formed by connecting all dendritic terminals of the fields by using Fiji/ImageJ (NIH, Bethesda, MD).

## Statistical analysis

The data are presented as the means ± standard deviation, and statistical significance was determined via unpaired Student's t-test, one sample t-test, non-repeated measures ANOVA with a post hoc Bonferroni correction using Microsoft Excel (Microsoft). For quantification, at least three cells per larvae and five different animals per condition were analyzed. The statistical significance was set at $p < 0.05$.

## Supporting information

**S1 Fig. *miR-87* is required for dendrite regeneration.** Dendrite regeneration in wild-type control (WT) and *miR-87* knockout (*miR-87*) C4da neurons at the indicated time points. Scale bar = 100 μm. Genotypes: WT, *ppk-GFP*; *miR-87*, *ppk-GFP*; *miR-87^{KO}/miR-87^{KO}*. (PDF)

**S2 Fig. Total dendrite length during pruning.** Total dendrite length of wild-type control (WT) and *miR-87* overexpressing (*miR-87* o/e) C4da neurons at white pupa (WP), 12 hours after pupal formation (12 hr APF), and 24 hours after pupal formation (24 hr APF). Note that no obvious difference was observed in dendrite pruning processes between WT and *miR-87*o/ e neurons. Error bars indicate mean ± S.D; n = 15; n.s., not significant (unpaired t-test). Genotypes: WT, *ppk-Gal4, UAS-mCD8GFP*; *miR-87o/e*, *ppk-GAL4, UAS-mCD8GFP;+/+; UAS-miR-87/+*. (PDF)

**S3 Fig. *hnf4* and *br* are dispensable for dendrite regeneration defects in *miR-87* neurons.** (A) Overexpression of *hnf4* or *br* causes dendrite regeneration defects. Morphology of *hnf4* overexpressing (*hnf4* o/e) and *br* overexpressing (*br* o/e) dendrites at 72 hr APF. Scale bar = 100 μm. (B) Reduction of *hnf4* or *br* dosage causes no obvious rescue in dendrite regeneration defects in *miR-87* KO C4da neurons. Morphology of *miR-87* KO with reduction of one copy of *hnf4* (*miR-87, hnf4/ miR-87*) or *br* (*br/+; miR-87/ miR-87*) dendrites at 96 hrs APF. Scale bar = 100 μm. (C) Quantification of total dendrite length in control wild-type (WT), *miR-87* KO (*miR-87*), *miR-87* KO with reduction of one copy of *ttk69* (*miR-87; ttk69/+*), *miR-87* KO with reduction of one copy of *hnf4* (*miR-87, hnf4/miR-87*), *miR-87* KO with reduction of one copy of *br* (*br/+; miR-87*) dendrites at 96 hrs APF. n = 15, Error bars indicate mean ± S. D., *p < 0.01 (ANOVA with a post hoc Bonferroni correction). Genotypes: WT, *ppk-Gal4, UAS-mCD8GFP/+*; *miR-87*, *ppk-Gal4, UAS-mCD8GFP/+*; *miR-87^{KO}/ miR-87^{KO}*; *miR-87*; *ttk69/+, ppk-Gal4, UAS-mCD8GFP/+*; *miR-87^{KO}/ miR-87^{KO}*; *ttk^{1e11}/+*; *miR-87, hnf4/miR-87, ppk-Gal4, UAS-mCD8GFP/+*; *miR-87^{KO}, hnf4^{Δ33}/ miR-87^{KO}*; *br/+; miR-87/miR-87, ppk-Gal4, UAS-mCD8GFP/br^{npr-3}; miR-87^{KO}/ miR-87^{KO}*. (PDF)

**S4 Fig. Expression of *ttk69* 3'UTR sensor in C4da neurons is suppressed in the larvae-to-pupae transition.** (A) A schematic illustration of the *ttk69* 3'UTR sensor construct. (B) Regulated expression of *ttk69* 3'UTR sensor. Images show expression of the *ttk69* 3'UTR sensor in wild-type (WT) and *miR-87* KO mutant (*miR-87*) C4da neurons at 2nd instar larval (2L) and white pupal (WP) additionally expressing *ppk-CD4-tdTomato*. Scale bar = 20 μm. (C) Quantification of the *ttk69* 3'UTR sensor GFP expression in C4da neurons at the indicated developmental stages. The *ttk69* 3'UTR sensor fluorescence intensity was normalized to the control sensor fluorescence intensity. n = 25, Error bars indicate mean ± S.D, *p<0.001 (Student's *t*-test). Genotypes: *ppk-CD4-tdTomato*; *miR-87^{KO}/miR-87^{KO}*; *tub-GFP-ttk69 3'UTR /+*
(PDF)

**S5 Fig. *ttk69* overexpression causes dendrite regeneration defects after injury.** (A, B) Overexpression of *ttk69* causes dendrite regeneration defects in adult. Morphology of wild-type control (WT) and *ttk69* overexpressing (*ttk69* o/e) dendrites at adult 1 day (A). Scale bar = 100 μm. Quantification of total dendrite length in wild-type (WT), *ttk69* overexpressing (*ttk69* o/e) neurons (B). n = 15, Error bars indicate mean ± S.D, *p<0.01 (Student's *t*-test). (C, D) Time-lapse images of dendrite regeneration after injury in wild-type (WT) and *ttk69*-overexpressed (*ttk69* o/e) neurons. Red arrows indicate the site of dendrite severing in 2nd instar larvae (44–48 hr AEL) (C). Scale bar = 50 μm. Quantification of regenerated dendrite length after severing in control wild-type (WT) and *ttk69*-overexpressed (*ttk69* o/e) neurons at 54 hr after severing. n = 6, Error bars indicate mean ± S.D, *p<0.05 (unpaired *t*-test). Genotypes: WT, *ppk-Gal4*, *UAS-mCD8GFP*; *ttk69 o/e*, *ppk-GAL4*, *UAS-mCD8GFP*; +/+; *UAS-ttk69/+*.
(PDF)

**S1 Table. miRNAs KO stocks screened in this study.** To visualize C4da dendrites, we introduced *ppk-GFP* reporter into miRNAs KO stocks. C4da dendrites of miRNAs KO homozygote animals were observed at 24, 48, 72 hr APF. Five C4da neurons from at least 3 independent animals were observed.
(TIFF)

**S2 Table. C4da dendrite phenotypes at 72 hr APF by overexpressing *miR-87*.** Potential *miR-87* targets were listed using the target prediction program (Stark et al. 2003). Among the top 100 potential candidates, UAS stocks were available for 43 candidates. The potential target genes were driven by the C4da neuron-specific *ppk-GAL4*. Dendrite phenotypes were observed at 72 hr APF.
(TIFF)

**S1 Movie. Time-lapse imaging of dendrite regeneration in wild-type control C4da neurons starting at 24 hr APF.**
(MOV)

**S2 Movie. Time-lapse imaging of dendrite regeneration in *miR-87* KO C4da neurons starting at 24 hr APF.**
(MOV)

## Acknowledgments

We would like to thank D. Anderson, T. Kitamoto, G. Rubin, W.M. Deng, Bloomington Stock Center, and Kyoto Stock Center for fly stocks; the member of the Emoto and Parrish labs for critical comments and discussion; M. Miyahara and H. Itoh for technical assistance.

## Author Contributions

**Conceptualization:** Yasuko Kitatani, Akane Tezuka, Jay Z. Parrish, Kazuo Emoto.

**Data curation:** Akane Tezuka, Eri Hasegawa, Masato Tsuji, Jay Z. Parrish.

**Formal analysis:** Yasuko Kitatani, Eri Hasegawa, Masato Tsuji, Jay Z. Parrish.

**Funding acquisition:** Jay Z. Parrish, Kazuo Emoto.

**Investigation:** Yasuko Kitatani, Akane Tezuka, Eri Hasegawa, Satoyoshi Yanagi, Kazuya Togashi, Jay Z. Parrish.

**Methodology:** Yasuko Kitatani, Akane Tezuka, Eri Hasegawa, Jay Z. Parrish.

**Resources:** Shu Kondo.

**Supervision:** Jay Z. Parrish, Kazuo Emoto.

**Validation:** Yasuko Kitatani, Eri Hasegawa, Masato Tsuji, Jay Z. Parrish.

**Visualization:** Yasuko Kitatani, Akane Tezuka, Eri Hasegawa.

**Writing – original draft:** Yasuko Kitatani, Akane Tezuka, Eri Hasegawa.

**Writing – review & editing:** Jay Z. Parrish, Kazuo Emoto.

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
