## [Decision Letter · Decision Letter 0]

30 Mar 2020

Dear Kazuo,

Thank you very much for submitting your Research Article entitled 'Drosophila miR-87 promotes dendrite regeneration by targeting the transcriptional repressor Tramtrack69' to PLOS Genetics. Your manuscript was fully evaluated at the editorial level and by independent peer reviewers. The reviewers appreciated the attention to an important problem, but raised some substantial concerns about the current manuscript. Based on the reviews, we will not be able to accept this version of the manuscript, but we would be willing to review again a much-revised version. We cannot, of course, promise publication at that time.

Should you decide to revise the manuscript for further consideration here, your revisions should address the specific points made by each reviewer. Particularly, Reviewer 3 commented that additional evidence is needed to support the conclusion that developmental and injury-induced regeneration share a common intrinsic mechanism, all reviewers asked you to clarify the roles of nf4 and br in the process, and Reviewer 1 suggested additional experiments on the GFP sensor. The reviewers also noted that some control experiments are missing. We will also require a detailed list of your responses to the review comments and a description of the changes you have made in the manuscript.

If you decide to revise the manuscript for further consideration at PLOS Genetics, please aim to resubmit within the next 60 days, unless it will take extra time to address the concerns of the reviewers, in which case we would appreciate an expected resubmission date by email to plosgenetics@plos.org.

[LINK]

We are sorry that we cannot be more positive about your manuscript at this stage. Please do not hesitate to contact us if you have any concerns or questions.

Yours sincerely,

Bing Ye

Guest Editor

PLOS Genetics

Gregory Barsh

Editor-in-Chief

PLOS Genetics

Reviewer's Responses to Questions

**Comments to the Authors:**

Reviewer #1: Summary: Kitatani, et al. present an interesting study showing that the microRNA, miR-87, is required for adult-specific dendritic growth during pupal metamorphosis. Loss of miR-87 impairs dendrite regeneration whereas over-expression of miR-87 leads to early initiation of regeneration. Moreover, miR-87 acts in a cell-autonomous manner to regulate dendritic growth. The authors present evidence consistent with Tramtrack 69 (ttk69) being a functional target of miR-87 and reducing ttk69 expression in miR-87 KO neurons restores dendritic growth. Finally, the authors show that miR-87 is required for dendrite regeneration after acute injury.

Major comments

1.) Throughout the manuscript and figures the exact genotype of “wild-type” is not clear. This leads to uncertainty about whether proper controls were used for each experiment. Presumably the wild-type in Figures 1, 3 and 4 is a fly with the same genetic background as the miR-87 KO mutant. For Figures 5, 6, and 7 it is not clear if the wild-type control is a GAL4-only control, a UAS-only control, or some other type of genetic background control. Making the genotype for each control explicitly clear is essential.

2.) Figure 4 - The authors should examine fluorescence from the control GFP sensor in miR-87 KO flies during 2L and WP stages to see if there are uniform levels of GFP expression in C4da neurons. Further, the methods used by the authors to quantify fluorescence in Figure 4H are unclear. The authors should quantify fluorescence of the miR-87 GFP sensor in WT and miR-87 KO neurons and normalize the data to fluorescence intensity of the control GFP sensor lacking miR-87 binding sites. This would demonstrate additional confidence that control of the miR-87 GFP sensor is dependent on miR-87 activity.

3.) Figure 6 - The authors indicate that over-expression of three miR-87 targets, ttk69, hnf4, and br, caused significant defects in dendrite regeneration. They follow up with dosage-sensitive interactions between the candidate genes and miR-87. However, only data for ttk69 is provided. Similar data for hnf and br should be shown. Specifically, assaying whether heterozygous loss-of-function mutations in hnf and br could suppress the regeneration defect in miR-87 KO neurons. If all three candidate genes show similar levels of suppression, the manuscript should specifically address why ttk69 was chosen for follow-up experiments. If the loss-of-function studies were inconclusive, those data should be included.

4) The supplemental videos are helpful, but I was struck by how the dendrites in the mutants appeared to fasciculate more than the wild-type. Given that these dendrites typically exhibit self-avoidance, in order to tile in the animal, could the authors comment on whether that is an observation that was made in the other live imaging studies? Is there a way to measure number of contacts observed? I do not need the authors to add additional data, commenting on this possibility would be sufficient, but if they have that data tabulated, adding it would be good.

5) The authors provide compelling evidence that ttk69 interacts genetically, and evidence that levels of ttk69 are elevated in miR-87 mutants. A complete proof that miR-87 regulates ttk69 post-transcriptionally would likely necessitate manipulating the 3’UTR binding sites in ttk69 and rescuing that by mutating miR-87 to recover binding and regulation. Thus, I would ask the authors to simply mention in the discussion that their evidence is consistent with the 3’ UTR regulation, but that there are potentially other factors that miR-87 could be regulating upstream of ttk69.

Minor comments

1.) Figure 1- Ideally, the authors should show representative images for dendrites at 48-hr APF and in 2-day and 3-day-old adults. Perhaps it would be easier to break this figure up into two figures, one for dendritic growth after puparium formation and a second figure for dendritic growth post-eclosion.

2.) p.8, 2nd paragraph - AFP should be APF

3.) p.10, 1st paragraph. Figure call-out error “…by the miR-87 sensor beginning in 3rd instar larvae, resulting in a >50% reduction of GFP expression in early pupal stages (Fig 4B-4G). However, 4B and 4C have control sensor data.

4.) p.15, 2nd paragraph - “…suggesting that the increased expression of miR-87…”. Expression of miR-87 was never demonstrated, only increased activity through the GFP sensor.

5.) In Figures 5, 6, and 7 the genotypes of each fly line are not clear. It would be helpful to write out the genotypes in the figure legend similar to the genotypes of MARCM clones in Figure 2.

“Clone genotypes: (WT) hsFLP, ppk-Gal4, UASmCD8GFP/+; FRT40A, (miR-87) hsFLP, ppk-Gal4, UAS-mCD8GFP/+; miR-87, FRT40A, and (miR-87 + UAS-miR-87) hsFLP, ppk-Gal4, UAS-mCD8GFP/+; miR-87, FRT40A; UAS-miR-87/+.”

5.) A section in the Methods should be added for quantification of GFP fluorescence in Figures 4 and 7.

6.) The authors quantify dendritic field size in Figure 2, but not in any other experiments. I did not see a rationale for this.

7.) p.30, Figure 3 legend - “Time stumps” should be “Time stamps”

8.) Could the authors please comment on when the pickpocket enhancer becomes active in the C4da neurons?

9.) Could the authors please confirm that the total dendritic length is the sum of all the dendrites from a single neuron. Or is it from multiple neurons within the same animal?

Reviewer #2: This study has demonstrated that microRNA miR-87 contributes to neuronal dendrite outgrowth in both developmental and injury-induced regeneration and has shown its underlying molecular mechanism at least at the level of identifying a critical target of miR-87, ttk69. All of the experiments were done in scientifically sound manners and the data are appropriately presented.

Minor comments:

1. What is the role of the miR-87 target, ttk69, in dendrite morphogenesis and regeneration of C4da neurons? The authors state roles of ttk69 in C1da neurons and R7 photoreceptor. If the role of ttk69 in C4da neurons is not known, state as such.

2. The positive hits of the primary screening of the miR-67 targets were hnf4, br and ttk69 (Figure 6A and 6B) and the authors examined ttk69 in their secondary test (Figure 6C and 6D). What happened to hnf4 and br? If they were negative in the secondary test or if the authors did not examine those candidates further, state as such.

3. It is preferable to describe the clone genotype in Figure 5 legend (did I overlook at it?)

Reviewer #3: Drosophila miR-87 promotes dendrite regeneration by targeting the transcriptional repressor Tramtrack69 (PGENETICS-D-20-00327)

In this manuscript the authors provide genetic evidence to report that microRNA miR-87 is required not only for the developmentally programed dendrite regeneration of Drosophila sensory neurons during metamorphosis, but also for the injury-induced dendrite regeneration of larval neurons. By monitoring the miR-87 sensor, they further show that the activity of miR-87 is upregulated in neurons during larvae-to-pupae transition, prior to dendrite regeneration, and also in larval neurons after dendrite injury. Finally, they identify one of the miR-87 targets, ttk69, and show that miR-87-mediated reduction of ttk69 is required to promote both the developmentally programed and the injury-induced dendrite regeneration.

The genetic studies of this manuscript are well designed and carried out. The miR-87 activity reported by the miR-87 sensor is nicely shown correlated with the ability of dendrite regeneration in C4da neurons, and with the reduction of ttk69 expression in neurons. The manuscript would need some improvement for it to be accepted by the journal. Here are the two main issues that I would like to raise for this manuscript.

First, the authors have done a lot for the developmentally programed dendrite regeneration; however, less has been done for the injury-induced dendrite regeneration of larval neurons. To support their main conclusion that a common intrinsic mechanism, ttk69 downregulation by miR-87 to promote dendrite growth in C4da neurons, is shared by both cases, the authors need to provide more evidence for the dendrite regeneration in larval neurons after injury.

1) In Fig 5, the authors show that miR-87 overexpression could cause precocious dendrite growth in C4da neurons at 30 hr APF. Does miR-87 overexpression also cause precocious dendrite growth in larval C4da neurons after injury?

2) In Fig 6, the authors show that ttk69 overexpression causes dendrite regeneration defects in C4da neurons at 72 hr APF. Does ttk69 overexpression also cause impaired dendrite growth in adult C4da neurons, like the ones in miR-87 KO adults (Fig 1)? Does ttk69 overexpression also cause impaired dendrite growth in larval C4da neurons after injury?

Second, the current results provided in this manuscript only suggest that ttk69 is likely one of the miR-87 target genes for dendrite regeneration. Although the authors show in Fig 6 that no decrease of ttk69 mRNA in the brain extracts of miR-87 KO animals at white pupal stages, this result still indirectly suggest that ttk69 is one miR-87 target in the brain. If the authors could provide evidence to demonstrate that ttk69 is the real target of miR-87 in C4da neurons, it would greatly strengthen the mechanistic insights into this manuscript.

Other points:

1) Do the authors try to rescue dendrite regeneration defects in miR-87 KO neurons with removal of one copy of hnf4 or br, or with RNAi of either genes? Or any combination of these three candidate genes reduction to see whether it can improve the rescue efficiency of dendrite regeneration defects in miR-87 KO neurons by single gene reduction?

2) On page 9, the cited information in the sentence, “A previous northern blot analysis has shown that miR-87 expression in the brain ……”, is not quite correct, according to ref 40. They used RNA from the whole animals, not the brain, at various developmental stages for northern blot analysis.

3) In the Discussion, the authors discuss the possible mechanism of temporal regulation of miR-87 expression dynamics, and think ecdysone might control the temporal expression of miR-87. However, according to the information in ref 40, the experimental manipulation of ecdysone or Juvenile hormone signaling in S2 cells (done by the authors of ref 40) does not affect the expression of miR-87. I think the authors should include this information and discuss other possibility that could regulate miR-87 expression temporally.

4) The results shown in Fig 2 demonstrated that dendrite regeneration defects were largely, not fully, rescued by UAS-miR-87 expression in the miR-87 MARCM clones of C4da neurons, suggesting that mir-87 has a role acting cell-autonomously in neurons for dendrite regeneration. Since it is not fully rescued by UAS-miR-87 expression in miR-87 MARCM clones, compared to the wild-type clones, it may suggest a possibility that miR-87 might also have a cell-non-autonomous role in dendrite regeneration. Have the authors done any experiments to rule out this possibility? Or can the authors discuss it in the Discussion?

**Have all data underlying the figures and results presented in the manuscript been provided?**

Reviewer #1: Yes

Reviewer #2: Yes

Reviewer #3: Yes

PLOS authors have the option to publish the peer review history of their article (what does this mean?). If published, this will include your full peer review and any attached files.

Reviewer #1: No

Reviewer #2: No

Reviewer #3: No

---

## [Decision Letter · Decision Letter 1]

17 Jun 2020

Dear Dr Emoto,

We are pleased to inform you that your manuscript entitled "Drosophila miR-87 promotes dendrite regeneration by targeting the transcriptional repressor Tramtrack69" has been editorially accepted for publication in PLOS Genetics. Congratulations!

Yours sincerely,

Bing Ye

Guest Editor

PLOS Genetics

Gregory Barsh

Editor-in-Chief

PLOS Genetics

Comments from the reviewers (if applicable):

Reviewer 1 had several minor comments on writing styles and typos, which we would like the authors to address.

Reviewer's Responses to Questions

**Comments to the Authors:**

Reviewer #1: The authors have satisfactorily addressed the major comments from the first review.

Minor comments

1.) Line 201 - Figure callout should be Fig 4H.

2.) Line 216 - Figure callout should be Supplemental Fig S2

2.) Line 779 - Supplemental Figure S3 title does not match Figure S3 content

3.) Line 788 - ttk69+/- should read ttk69/+

4.) Line 807 - URT should read UTR

Reviewer #3: The revised version seems to be improved over the original paper. The authors have fully addressed my concerns. Therefore, I support its publication.

**Have all data underlying the figures and results presented in the manuscript been provided?**

Reviewer #1: Yes

Reviewer #3: Yes

PLOS authors have the option to publish the peer review history of their article (what does this mean?). If published, this will include your full peer review and any attached files.

Reviewer #1: No

Reviewer #3: No

**Data Deposition**

http://datadryad.org/submit?journalID=pgenetics&manu=PGENETICS-D-20-00327R1

**Press Queries**

---

## [Editor Report · Acceptance letter]

23 Jul 2020

PGENETICS-D-20-00327R1 

Drosophila miR-87 promotes dendrite regeneration by targeting the transcriptional repressor Tramtrack69 

Dear Dr Emoto, 

We are pleased to inform you that your manuscript entitled "Drosophila miR-87 promotes dendrite regeneration by targeting the transcriptional repressor Tramtrack69" has been formally accepted for publication in PLOS Genetics! Your manuscript is now with our production department and you will be notified of the publication date in due course.

With kind regards,

Matt Lyles

PLOS Genetics

On behalf of:
